# Genome-Wide Selective Signature Analysis Revealed Insecticide Resistance Mechanisms in *Cydia pomonella*

**DOI:** 10.3390/insects13010002

**Published:** 2021-12-21

**Authors:** Wen-Ting Dai, Jin Li, Li-Ping Ban

**Affiliations:** Department of Grassland Resources and Ecology, College of Grassland Science and Technology, China Agricultural University, Beijing 100193, China; Daiwenting0513@163.com (W.-T.D.); lijin2017@163.com (J.L.)

**Keywords:** codling moth, insecticide, resistance, selective sweep

## Abstract

**Simple Summary:**

The codling moth, *Cydia pomonella*, is a quarantine pest that causes extensive damage to many important pome fruits. To control this pest, insecticides are frequently used, leading to the development of resistance. In this study, we analyzed resequencing data of two resistant and one susceptible strains of codling moth, detecting the positively selected genes under the insecticide selective pressure. Coupled with transcriptome data, we discussed the potential role in insecticide resistance of these positively selected genes. Our results identified eight genes including *CYP6b2*, *CYP307a1,* *5-hydroxytryptamine*
*receptor*, *cuticle protein*, and *acetylcholinesterase*, which are potentially involved in cross-resistance to azinphos-methyl and deltamethrin. Overall, our finding indicated that the insecticide resistance mechanism in *C. pomonella* is a complex physiological and biochemical process.

**Abstract:**

The codling moth, *Cydia pomonella* L. (Lepidoptera, Tortricidae), is a serious invasive pest of pome fruits. Currently, *C. pomonella* management mainly relies on the application of insecticides, which have driven the development of resistance in the insect. Understanding the genetic mechanisms of insecticide resistance is of great significance for developing new pest resistance management techniques and formulating effective resistance management strategies. Using existing genome resequencing data, we performed selective sweep analysis by comparing two resistant strains and one susceptible strain of the insect pest and identified seven genes, among which, two (glycine receptor and glutamate receptor) were under strong insecticide selection, suggesting their functional importance in insecticide resistance. We also found that eight genes including *CYP6B2*, *CYP307a1*, *5-hydroxytryptamine receptor*, *cuticle protein*, and *acetylcholinesterase*, are potentially involved in cross-resistance to azinphos-methyl and deltamethrin. Moreover, among several P450s identified as positively selected genes, *CYP6B2*, *CYP4C1*, and *CYP4d2* showed the highest expression level in larva compared to other stages tested, and *CYP6B2* also showed the highest expression level in midgut, supporting the roles they may play in insecticide metabolism. Our results provide several potential genes that can be studied further to advance understanding of complexity of insecticide resistance mechanisms in *C. pomonella*.

## 1. Introduction

Codling moth, *Cydia pomonella* L. (Lepidoptera, Tortricidae), is a serious insect pest of many economically important pome fruit trees including apples, pears, and walnuts [1]. The larvae damage fruit by boring into them, resulting in the reduction of yield and quality. Native in south-central Eurasia, this invasive species has now been found in all six continents except Antarctica aided by the widespread cultivation of apple trees [2]. Due to its severe damage on pome fruits, *C. pomonella* has been listed as a quarantine pest by a number of countries across the globe [3]. In China, its occurrence was reported in nine provinces despite a close monitoring being implemented [4]. 

Currently, the management of *C. pomonella* mostly relies on the application of chemical reagents [5]. However, prolonged and excessive use of chemical insecticides has led to serious resistance problems, making many insecticides less effective [6,7]. For example, in recent years *C. pomonella* has evolved resistance to several pyrethroids and organophosphates [8,9].

Three major types of insecticide resistance mechanisms have been described: (i) metabolic resistance that involves overexpression and elevated catalytic activity of detoxification enzymes, (ii) target resistance that involves mutation of the insecticide target site, and (iii) penetration resistance that involves modifications of the cuticle [10,11]. Owing to the excessive use of chemical insecticides for *C. pomonella* control, selection pressure has been imposed on the evolution of insecticide resistance in this insect pest, making it is an excellent model species to decipher the molecular mechanisms of insecticide resistance. 

Under selection pressure, favorable mutation occurs and fixes in a population. Linked neutral mutations then ‘hitchhike’ to fix with the favorable mutation. This ‘hitchhiking effect’ will cause reduced diversity [12], increased linkage disequilibrium and reduced heterozygosity around the selected locus [13], a so-called ‘selective sweep’. Currently, various methods are available for selective sweep analyses to detect genomic regions associated with phenotype traits. For example, genomic regions affected by prolonged DDT selection in *Drosophila melanogaster* [14] and genes associated with pyrthroid and DDT resistance in *Amyelois transitella* have been reported [15]. These successful examples indicate that selective sweep analyses are feasible for insecticide resistance studies, supported by high-quality genome sequence assemblies, model systems of insecticide-resistant insects, and tools for genome-wide molecular analyses.

In this study, we analyzed genome resequencing and transcriptomic data of three *C. pomonella* strains, comprising one stain susceptible to both azinphos-methyl and deltamethrin, one resistant to azinphos-methyl only, and one resistant to deltamethrin only [16]. We identified several selection signatures and discussed the roles they may play in insecticide resistance from the perspectives of adaptive evolution and population genetics.

## 2. Materials and Methods

### 2.1. C. pomonella Genome Resequencing and Transcriptomic Data

The *C. pomonella* genome resequencing and transcriptomic Sequence Read Archive (SRA) data were downloaded from the National Center for Biotechnology Information (NCBI). The genomic data comprised 18 samples of three insect strains (one susceptible strain, ‘S’, and two resistant strains, ‘Raz’ and ‘Rde’), with each strain containing six samples (NCBI accession: SRR8479443-SRR8479460). The ‘Raz’ strain comes from Lerida, Spain. The ‘Rde’ and ‘S’ strains are from south-eastern France. The Raz strain has been selected for insecticide resistance by exposing larvae to azinphos-methyl, and it shows 7-fold resistance to azinphos-methyl in comparison with the S strain. The Rde strain has been selected by exposing larvae to deltamethrin and showed 140-fold resistance to deltamethrin in comparison with the S strain. *Cydia pomonella* RNA-seq data were obtained and analyzed for the different developmental stages (egg, pupa, larva, and adult; NCBI accession: SRR8479433-SRR8479442), and tissues (accessory gland, head, midgut, ovary, and testis; NCBI accession: SRR4101328-SRR4101341) [16]. 

### 2.2. Read Alignment and Variant Calling

The sequence reads were filtered using NGS QC Tool kit (v2.3.3) with default parameters to remove the low-quality ones [17]. The obtained clean data were aligned using BWA-MEM (v0.7.15) to the *C. pomonella* reference genome (http://www.insect-genome.com/cydia/ (accessed on 12 December 2021), *Cydia pomonella* genome chromosomes v1). Sequence Alignment/Map (SAM) format files were sorted with the Picard tools SortSam (v2.2.4) and then converted to Binary sequence Alignment/Map (BAM) format files. Duplicate reads were removed from each sample alignment using the Picard tools MarkDuplicates (v2.2.4). 

Prior to SNP calling, Genome Analysis ToolKit (v3.6), Realigner Target Creator, and Indel Realigner were used for global realignment. SNPs were called using GATK UnifiedGenotyper with the min_base_quality_score of 20, stand_call_conf of 30 and stand_emit_conf of 30. GATK VariantFiltration was subsequently used to remove the unconfident variant sites with the setting of QUAL < 30.0, QD < 5.0, FS > 60.0, MQ < 40.0, MQRankSum < −12.5, and ReadPosRankSum < −8.0.

### 2.3. Population Genetics Analysis

A phylogenetic tree was constructed with IQ-TREE (v1.6.12) using the maximum likelihood method. iTOL (https://itol.embl.de/ (accessed on 12 December 2021)) was used to visualize the phylogenetic tree. Principal component analysis (PCA) of all SNPs was performed using the PLINK (v1.90b6.4). All SNPs were divided into 24 datasets and each SNP dataset was used for clustering analysis using the program ADMIXTURE (v1.3.0). Plots were constructed using the library ggplot2 of R.

### 2.4. Selective Sweep Analysis

To detect positively selected genes (PSGs) related to insecticide resistance, we calculated the population differentiation index (F_ST_), nucleotide diversity (θπ) ratio and the Tajima’s D value [18,19,20]. F_ST_ and θπ were calculated with VCFtools (v0.1.13) using a 5 kb window with a 1 kb step. The negative and missing F_ST_ values were discarded, because these values have no biological interpretation [21]. The θπ ratio was calculated as θπ(susceptible)/θπ(resistant). Tajima’s D was calculated with VCFtools (v0.1.13) using a 5 kb window.

### 2.5. Quantitative Analysis of Gene Expression Levels in Different Tissues and Stages

We used the fastp (v0.20.0) to filter out the low-quality reads and trim adapters with the default parameters [22]. After building a HISAT2 index using hisat2-build, the clean reads were mapped to the *C. pomonella* reference genome using HISAT2 (v2.1.0) [23]. The FPKM value of each gene was determined using Stringtie (v2.1.4) based on the annotated *C. pomonella* GFF file (http://www.insect-genome.com/cydia/ (accessed on 12 December 2021), *Cydia pomonella* OGS[GFF3] v1). 

## 3. Results

### 3.1. Genetic Differences in Resistance and Susceptible Strains

Genomic resequencing data of 18 samples belonging to three strains were mapped to the *C. pomonella* reference genome. A total of 1.45 million high-quality SNPs were detected among all samples. PCA revealed a clear split between resistant and susceptible strains (Figure 1a). The first and the second principal components (PCs) accounted for 18.83% and 15.83%, respectively, of the total variations separating the three populations. The phylogenetic tree and population structure yielded similar results (Appendix A). Furthermore, reduced genetic diversity was observed both in Rde (4.555 × 10^−3^ *t*-test, *p* < 2.22 × 10^−16^) and Raz (5.219 × 10^−3^, *t*-test, *p* < 2.22 × 10^−16^) strains in comparison with the susceptible strain (5.372 × 10^−3^) (Figure 1b), suggesting that the resistance strains were under strong selection from continuous use of insecticides.

### 3.2. Insecticide-Related Genes Detected by Selective Sweep

We searched the *C. pomonella* genome regions with the top 5% F_ST_ to detect signatures of positive selection, and subsequently found 784 and 809 genes from the Raz and Rde strains, respectively (Appendix A). Among the 137 PSGs showing strong selective signatures common to both resistant strains, eight appeared to be involved in insecticide resistance as indicated previously, including genes of cation channel, P450, acetylcholinesterase, cuticle protein, and 5-hydroxytryptamine receptor (Figure 2, Table 1). Furthermore, the Tajima’s D of them deviated from 0 in resistant populations, indicating that they were under selection pressure (Appendix A). As their selective signatures were detected in both resistant strains, these genes could be involved in the development of resistance to both azinphos-methyl and deltamethrin.

We compared the candidate insecticide-resistance genes identified in this study and those previously detected using the GWAS (Genome-Wide Association Studies) approach and found 21 PSGs which were not detected in the GWAS analysis [16]. These 21 PSGs, including genes of chitinase protein, gamma-aminobutyric acid receptor, ATP-binding cassette transporters, glutamate receptor, voltage gated calcium channel, cytochrome P450, acetylcholine receptor, glycine receptor, and glutathione S-transferase (Table 2), are likely important insecticide resistance genes, because they may act as insecticide targets, be involved in insecticide detoxification, or contribute to the alteration of insecticide penetration. Most PSGs were expressed in larval, except glutamate receptor mainly in eggs, glycine receptor, and CYP4g15 in pupa (Figure 3a). Moreover, genes of detoxifying enzymes, such as ATP-binding cassette transporters and glutathione S-transferase, showed highest expression level in midgut, while genes of target receptors, such as gamma-aminobutyric acid receptor, glutamate receptor, acetylcholine receptor and glycine receptor, as well as calcium channel were mainly expressed in head of *C. pomonella* (Figure 3b).

To predict the candidate insecticide resistance genes with highest confidence, we selected the PSGs with highest 5% F_ST_ and θπ (susceptible/resistant) values for further analysis, including 431 from the Raz strain and 424 from the Rde strain (Appendix A; Appendix A)). Interestingly, of the 21 PSGs that were not detected previously using the GWAS analysis, seven fell in the range with the highest nucleotide diversity ratio (top 5%) (Table 3; Figure 4 and Appendix A), and Tajima’s D deviated from 0 was also observed in Raz or Rde strains (Appendix A). In particular, CPOM07387 (glycine receptor subunit alpha-2) and CPOM14990 (glutamate receptor 1) showed both high F_ST_ (CPOM07387: 0.80; CPOM14990: 0.87) and θπ ratio (CPOM07387: 7.40; CPOM14990: 4.88) compared to neighboring regions. Their roles were also confirmed by lower values of Tajima’s D in Raz and Rde strains, respectively (Appendix A). In addition, we detected 11 and 2 homozygous SNPs in the Raz and Rde strains respectively, which were absent in the susceptible strain (Figure 4). 

### 3.3. Selective Signature and Spatial-Temporal Expression Pattern of Cytochrome P450 Enzymes

Metabolism of insecticides by P450 enzymes is a key factor determining resistance in insects [24]. P450-dependent desulfuration and hydroxylations are believed to be involved in the metabolism of organophosphorus and pyrethroid pesticides, which can lead to insecticide resistance [25]. Given the importance of P450s in insecticide resistance, we characterized the PSGs of P450 in this study. Five and seven P450 PSGs were detected in the Raz and Rde strains, respectively (Appendix A; Figure 5). CYP (cytochrome P450) genes in insects are composed of four clans, i.e., the CYP2, CYP3, CYP4, and mitochondrial CYP clans [25]. Considerable evidence links members of the CYP3 clan and CYP4 clan, especially CYP6s, CYP9s, and CYP4s, with insecticide resistance [26,27,28,29,30]. In this study, we also found that the positively selected P450s are mainly CYP6s, CYP9s, and CYP4s (Appendix A), suggesting their roles in insecticide resistance. Of interest, *CYP307a1* (geneID: CPOM09450) and *CYP6B2* (geneID: CPOM05212) showed high F_ST_ in both Raz and Rde strains (Figure 5, Table 1). We also found that 11 SNPs of *CYP307a1* showed genotype differences between the resistant and susceptible strains (Appendix A).

To investigate how these positively selected P450s were expressed across *C. pomonella* tissue types and life stages, we performed expression analysis using the RNA-Seq data of different tissues types (accessory gland, head, midgut, ovary, and testis) and life stages (egg, pupa, larva, and adult) (Figure 6). In comparison to other tissues, *CYP6B2* (geneID: CPOM03544) was expressed at the highest level in the midgut (Figure 6). Given that midgut is the important interface for food digestion and insecticide detoxification, this P450 gene may function as insecticide degrading molecules conferring insecticide resistance to the insect. In comparison to other life stages, three P450 genes, i.e., *CYP6B2 (*geneID: CPOM05212), *CYP4C3* (geneID: CPOM08186), and *CYP4C1* (geneID: CPOM18543), showed the highest expression in larva, the most feeding-active life stages, suggesting their possible roles in insecticide metabolism.

## 4. Discussion

Azinphos-methyl and deltamethrin are neurotoxins belonging to the organophosphates and pyrethroids classes, respectively. Deltamethrin targets primarily the voltage-gated sodium channels (VGSCs), where prolonged opening of Na^+^ channels, persistent depolarization, and repetitive firing lead to seizure, paralysis, and death of insects [31,32]. By contrast, the toxicity of organophosphates is attributed to their inhibition of insect acetylcholinesterase (AChE), an enzyme catalyzing the hydrolysis of acetylcholine (Ach) at the synaptic regions of cholinergic nerve endings. Inhibition of AchE leads to a buildup of Ach in the synapse and causes cholinergic neuronal excitotoxicity and dysfunction [33,34]. In the present study, we detected eight resistance-related genes (PSGs) which displayed strong selective signatures in the two resistant strains of *C. pomonella*. These PSGs are mainly P450s, cuticle proteins, and target receptors (Figure 2b; Table 1), making them possibly involved in conferring cross-resistance to both organophosphorus and pyrethroid pesticides. Cross-resistance refers to that resistance to one particular insecticide may cause resistance to other insecticides because of the same resistant mechanism of insect and the action mechanism of insecticides [35]. Properties of P450 enzymes with broad substrates could confer insecticide cross-resistance. Indeed, studies have indicated that a single P450 enzyme can metabolize a variety of insecticides. For example, *Anopheles* CYP6P3 was shown to be able to metabolize both bendiocarb and pyriproxyfen chemicals [36,37], and several other *Anopheles* P450s associated with pyrethroid-resistance also showed the metabolic capacity to other classes of insecticides, such as organophosphates [38]. P450-dependent desulfuration and hydroxylations involving the metabolism of organophosphorus and pyrethroid insecticides may account for the mechanism of cross-resistance to these two classes [25]. Our study detected two P450s—*CYP307a1* and *CPY6B2*—which could contribute to the development of cross-resistance to azinphos-methyl and deltamethrin in *C. pomonella*. *CYP6B2* has been indicated to be involved in the resistance to deltamethrin or azinphos-methyl [16], and *CYP307a1* is thought to participate in the biosynthesis of ecdysone [39,40]. Despite the fact that the role of *CYP307a1* as a detoxication gene to confer insecticide resistance remains to be determined, it has been suggested to cause imidacloprid resistance in *Sitobion avenae* [41]. In addition, we consider that the cuticle protein genes detected in our selective sweep analysis could confer cross-resistance to azinphos-methyl and deltamethrin insecticides owing to the roles they play in forming insect epidermis. The cuticle is the first physical barrier to prevent the entry of foreign materials such as pesticides, consisting of epicuticle and procuticle [10]. The epicuticle is mainly composed of hydrocarbons and lipids [42], and the procuticle is composed of chitin fibers and cuticle proteins [43]. Insect cuticle thickness was considered to be correlated with insecticide resistance as a thick cuticle layer may reduce insecticide penetration [44,45]. In *Drosophila*, chitin layer thickening was considered to contribute to the development of penetration resistance to drugs [46]. Because cuticle proteins are important components of cuticles, they are crucial in determining cuticle thickness and therefore may have influence on insecticide resistance in insects. For example, the knockdown of some cuticle protein genes in *Nilaparvata lugens* led to the reduction of procuticle thickness [47], and the expression of these cuticle protein genes were upregulated in the resistant to certain pesticides [48]. Because both azinphos-methyl and deltamethrin are applied via spraying and function through penetrating insect epidermis, it is conceivable that the cuticle proteins genes detected in our analysis may play a role in thickening cuticles to prevent insecticide entry and thus contribute to the cross resistance to these two insecticides. 

Natural or artificial selection of favorable mutations leads to reduced polymorphism and increased LD (linkage disequilibrium) and allele frequency [12,49,50]. Accordingly, identification of selective signatures is usually based on (i) population differentiation, such as F_ST_, by comparison of allele frequencies among different subgroups; (ii) increased LD, such as XP-EHH, by comparison of haplotype homozygosity among different subgroups; and (iii) nucleotide polymorphism, with a lower level of polymorphism indicating stronger selection. Application of at least two of these features is an effective strategy to predict strong selective genomic signatures with reduced false positive rates. In the present study, we used the F_ST_ and θπ ratio (θπ(susceptible/resistant)) approaches to detect particularly strong signs of selective sweeps presumably associated with insecticide-resistance and identified seven resistance-related genes which were not detected previously using the GWAS approach [16]. Among these genes, CPOM14990 (Glutamate receptor 1) and CPOM07387 (Glycine receptor subunit alpha-2) showed strong selective signature, which may be suggestive of their functional importance (Figure 4). Both glutamate receptors and glycine receptors are cysteine-loop ligand-gated ion channel proteins, functioning in nerve signal transmission [51,52] and having been shown to serve as drug targets [53,54,55]. Besides, of the seven PSGs, 5-hydroxytryptamine receptor, and gamma-aminobutyric acid type B receptor also play important role in nerve conduction (Table 3). Indeed, nervous system sensitivity to insecticides of resistant insects has shown decline compared to susceptible insects, which were thought to be related with the target receptor insensitivity. However, the insect nervous system is a complex mechanism and insecticides may not be the only target. For example, organophosphorus can also act on Ach receptors except for AChE [56], and pyrethroids act on voltage-gated calcium channels, gamma-aminobutyric acid receptors, and glutamic acid receptors, except for the Na^+^ channels [31,57,58]. Therefore, these genes associated with nerve conduction may have played a crucial role in adaptive evolution to insecticides by regulating the insect nerve system, as a result of long-term selection by neurotoxic insecticides. 

Note that though the candidate genes listed in Table 1 and Table 2 can be connected to insecticide resistance, many other PSGs are also detected in this study (Appendix A). However, their function is at this stage unknown, so their potential role in insecticide resistance is unknown. These results reported here are similar to other insecticide resistance studies where selective sweep analyses has been used [14,15]. Moreover, cytochrome P450s are always detected in selective sweep analyses, similar to our results. This indicates that they may be the main targets for selection and have functional importance in insecticide resistance. 

## Figures and Tables

**Figure 1 insects-13-00002-f001:**
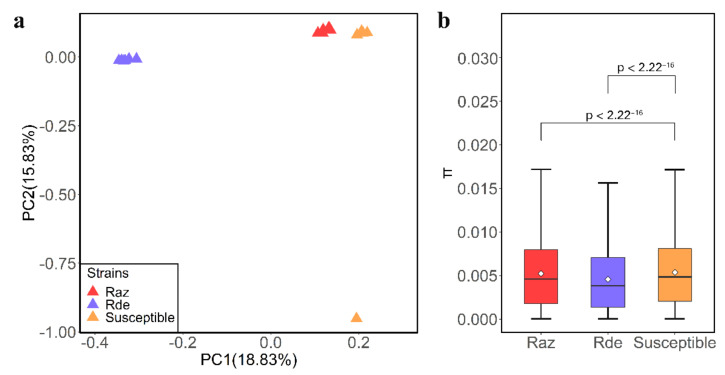
Genetic differentiation between susceptible and resistant (Raz and Rde) strains of *C. pomonella.* (**a**) Principal component analysis (PCA) of SNPs. The red and purple triangles represent the samples resistant to azinphos-methyl (Raz) and deltamethrin (Rde), and orange triangles represent the samples susceptible to azinphos-methyl and deltamethrin (S). (**b**) Boxplot showing the population nucleotide diversity (θπ) of the three stains of *C. pomonella* (*t*-test, *p* < 2.22 × 10^−16^), indicating minimum, low quartile, median, mean, high quartile, and maximum values.

**Figure 2 insects-13-00002-f002:**
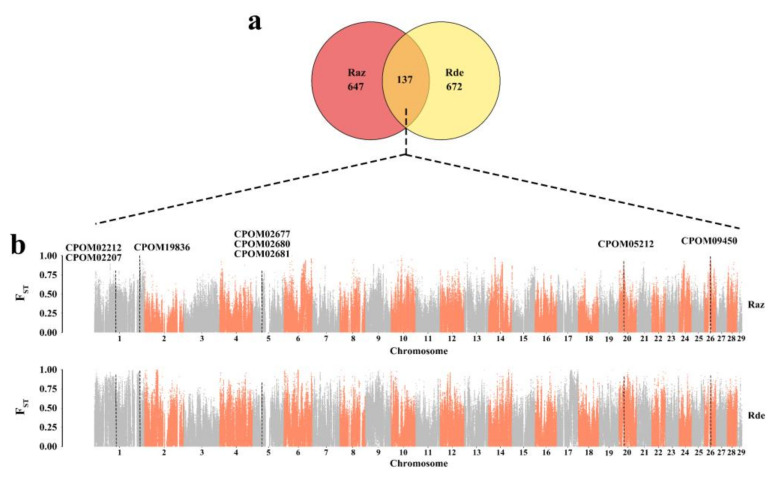
Location in genome and F_ST_ value of eight PSGs detected both in Raz and Rde strains of *C. pomonella*, respectively. (**a**) Venn diagram illustrating the number of common and unique PSGs detected in the Raz and Rde strains. (**b**) F_ST_ values of the eight resistant-related genes involved in cross-resistance. X-axes represent the location of genome, and orange and grey areas represent the chromosomes 1–29. Dotted lines parallel to the *y*-axis show the location in genome of eight PSGs. The Specific F_ST_ values were showed in Table 1.

**Figure 3 insects-13-00002-f003:**
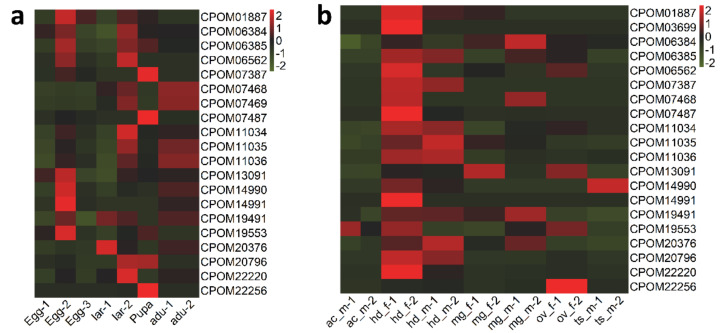
Expression pattern of 21 PSGs in different stages (**a**) and different tissue (**b**) of *C. pomonella*. The *x*-axis lar represents the different tissues and stages, and *y*-axis shows the PSGs. The color of grid represents the expression level of the PSGs in different tissues and stages. lar: larva; adu: adult; ac: accessory gland; hd: head; mg: midgut; ov: ovary; ts: testis; m: male; f: female.

**Figure 4 insects-13-00002-f004:**
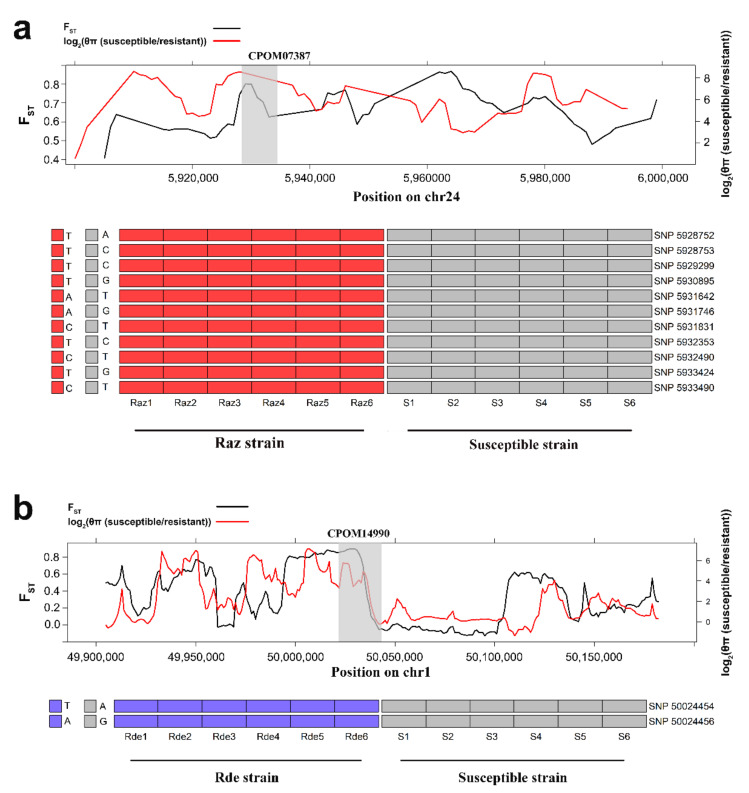
(**a**) *Glycine receptor* (geneID: CPOM07837) and (**b**) *glutamate receptor* (geneID: CPOM14990) showing different genetic signatures between resistant and susceptible *C. pomonella* strains. The upper parts of the figure show the F_ST_ (red line) and θπ (susceptible/resistant) (black line) plot around *glycine receptor* and *glutamate receptor.* The *x*-axis represents the location in chromosomes (bp), and the gray area shows the location of *glycine receptor* and *glutamate receptor*. The lower part shows the 11 and 2 homozygous SNPs identified in the two resistant strains, which were absent in the susceptible strains. SNPs and INDELs were named according to their position on the chromosome. The red (or purple) grids represent the homozygous SNPs.

**Figure 5 insects-13-00002-f005:**
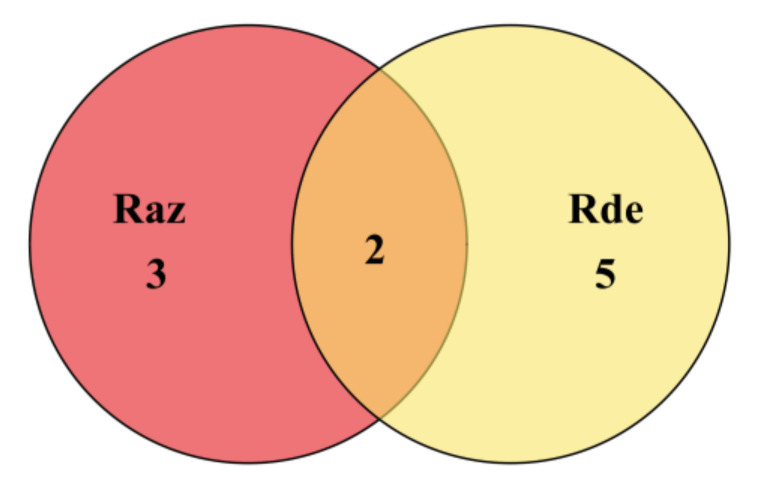
Venn diagram illustrating the the numbers of common and unique positively selected P450s in both Raz and Rde resistant strains of *C. pomonella*.

**Figure 6 insects-13-00002-f006:**
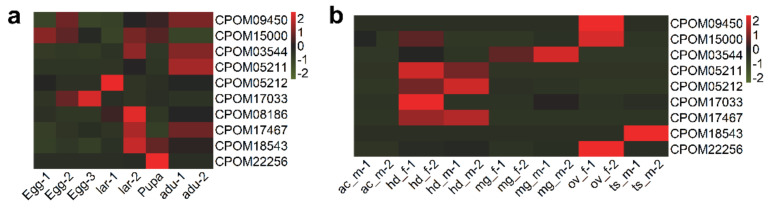
Expression pattern of positively selected P450s in different tissue (**a**) and different stages (**b**) of *C. pomonella*. The *x*-axis shows the different life stages (**a**) and tissues (**b**), and the *y*-axis shows the PSGs. The color of the grid represents the expression level of the PSGs in different tissues and stages. Key: lar = larva; adu = adult; ac = accessory gland; hd = head; mg = midgut; ov = ovary; ts = testis; m = male; f = female.

**Table 1 insects-13-00002-t001:** F_ST_ values of eight resistance-related genes showing strong selective signature in both Raz and Rde strains of *C. pomonella,* respectively. Maximum F_ST_ value is 1.00.

geneID	Raz_F_ST_	Rde_F_ST_	Name
CPOM19836	0.73	0.62	Transient receptor potential cation channel subfamily A member 1
CPOM05212	0.73	0.66	Cytochrome P450 6B2
CPOM09450	0.93	0.63	Cytochrome P450 307a1
CPOM02212	0.50	0.71	Acetylcholinesterase
CPOM02680	0.59	0.61	Cuticle protein 8
CPOM02681	0.58	0.77	Cuticle protein 19
CPOM02207	0.50	0.63	5-hydroxytryptamine receptor 2A
CPOM02677	0.63	0.66	Cuticle protein 19

**Table 2 insects-13-00002-t002:** F_ST_ values of 21 resistance-related PSGs detected in the Raz and Rde strains of *C. pomonella*, respectively.

geneID	Chromosome	F_ST_	Name
**Raz**			
CPOM07487	chr24	0.55	Glycine receptor subunit alpha-2
CPOM07387	chr24	0.80	Glycine receptor subunit alpha-2
CPOM06562	chr10	0.68	Metabotropic glutamate receptor
CPOM07469	chr24	0.56	Neuronal acetylcholine receptor subunit alpha-3
CPOM13091	chr12	0.81	ATP-binding cassette sub-family A member 3
CPOM06385	chr10	0.54	ATP-binding cassette sub-family B member 6, mitochondrial
CPOM06384	chr10	0.56	ATP-binding cassette sub-family B member 6, mitochondrial
CPOM19553	chr14	0.56	ATP-binding cassette sub-family G member 4
**Rde**			
CPOM03699	chr17	0.65	5-hydroxytryptamine receptor 2A
CPOM07468	chr24	0.60	Acetylcholine receptor subunit alpha-L1
CPOM19491	chr1	0.95	Chitinase-like protein EN03
CPOM22256	chr4	0.60	Cytochrome P450 4g15
CPOM22220	chr27	0.60	Cytochrome P450 6B2
CPOM01887	chr1	0.93	Gamma-aminobutyric acid type B receptor subunit 1
CPOM14991	chr1	0.70	Glutamate receptor 1
CPOM14990	chr1	0.87	Glutamate receptor 1
CPOM20796	chr4	0.50	Glycine receptor subunit alpha-1
CPOM20376	chr7	0.51	Microsomal glutathione S-transferase 1
CPOM11035	chr12	0.52	Voltage-dependent T-type calcium channel subunit alpha-1H
CPOM11034	chr12	0.71	Voltage-dependent T-type calcium channel subunit alpha-1I
CPOM11036	chr12	0.69	Voltage-dependent T-type calcium channel subunit alpha-1I

**Table 3 insects-13-00002-t003:** Seven PSGs detected with top 5% θπ ratio and top 5% F_ST_ in the Raz and Rde strains, respectively of *C. pomonella*.

geneID	Chromosome	F_ST_	θπ Ratio	Name
**Raz**				
CPOM07487	chr24	0.55	4.82	Glycine receptor subunit alpha-2
CPOM07387	chr24	0.80	7.40	Glycine receptor subunit alpha-2
**Rde**				
CPOM03699	chr17	0.65	6.03	5-hydroxytryptamine receptor 2A
CPOM01887	chr1	0.93	3.80	Gamma-aminobutyric acid type B receptor subunit 1
CPOM14991	chr1	0.70	6.48	Glutamate receptor 1
CPOM14990	chr1	0.87	4.88	Glutamate receptor 1
CPOM19491	chr1	0.95	5.71	Chitinase-like protein EN03

## Data Availability

The VCF file of SNPs is openly available in FigShare at https://doi.org/10.6084/m9.figshare.17209595. The F_ST_, π and Tajima’s D results are openly available in FigShare at https://doi.org/10.6084/m9.figshare.17206622. The expression data are available in Figshare at https://doi.org/10.6084/m9.figshare.17206574 (accessed on 12 December 2021).

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
