# Peer review of "Genome-Wide Selective Signature Analysis Revealed Insecticide Resistance Mechanisms in Cydia pomonella"

_insects, 2021, doi:10.3390/insects13010002_

Round 1

Reviewer 1 Report

The paper by Dai et al. reports genome-wide insecticide selection signatures in Cydia pomonella. The experiments are generally well-planned and results are well-presented. However, there are some issues that need to be addressed before making a decision on this paper.

Major issues that need to be addressed:

  1. The paper has too many grammatical errors and that makes it difficult for reviewers to accurately understand what the authors want to say or imply. Therefore, this paper needs to be reviewed for grammatical errors first and then submitted for second round of scientific peer-review.
  2. Although this paper has performed selective sweep analysis for identifying genes under positive selection pressure it does not provide enough background about the selective sweep analysis in the Introduction section (e.g., it does not mention previous selective sweep work done with other insect pests). Secondly, methods used for selective sweep analysis need to be mentioned in more detail and/or papers that describe the selective sweep analysis approach need to be cited. Lastly, authors need to discuss findings of other papers (with other insects) that performed insecticide resistance associated selective sweep analysis in relation to the findings of this paper.
  3. In terms of gene expression analysis using RNAseq data, I am wondering why the authors have not validated the RNAseq-based differential gene expression data using RT-qPCR? This is a common practice for all papers that report RNAseq-based differential gene expression data. Authors should at least test a sub-set of genes under positive selection for gene expression using the qPCR technique.

Reviewer 2 Report

In their article insects-1438670, the authors present a genomic study of pesticide resistant and susceptible strains of the important crop pest Cydia pomonella, based on the analysis of publicly available sequence and transcriptome datasets. These analyses identify a number of candidate genes that could be important for the development of insecticide resistance in this species. The manuscript is well written (but see attached PDF for some minor corrections throughout), and the results easy to interpret. I would suggest, however, that the authors limit the presentation of the candidate genes to only those with Fst or theta-pie values in the top 5% (as they do later in the manuscript). Using an arbitrary cutoff, such as Fst 0.5, confounds variation between the populations that could be the result of genetic drift, whereas the more stringent "outlier" approach would focus the results only on those that appear to be under selection. Though not necessary, the authors could also provide estimates for Tajima's D, as this test is easily performed in the VCFtools package they utilize, and might provide some additional insights.

A minor point. It would be helpful if the authors described where the samples are from.  Are the different resistant strains from different geographic localities? If so, further discussion about the known population structure of C. pomonella is needed.

Round 2

Reviewer 1 Report

Authors have made all of the suggested changes. However, grammatical errors are still present throughout the paper. Those errors need to be corrected before the formal acceptance of this paper.

Author Response

  1. Authors have made all of the suggested changes. However, grammatical errors are still present throughout the paper. Those errors need to be corrected before the formal acceptance of this paper.

Response 1: Thank you for spending time tin reviewing our manuscript and provide suggestion for improvement. And we made grammatical corrections to the manuscript with the help with an English native speaker. And the changes have been marked as red in the revised manuscript.